# Exploration of Bioengineered Scaffolds Composed of Thermo-Responsive Polymers for Drug Delivery in Wound Healing

**DOI:** 10.3390/ijms22031408

**Published:** 2021-01-30

**Authors:** Luis Castillo-Henríquez, Jose Castro-Alpízar, Mary Lopretti-Correa, José Vega-Baudrit

**Affiliations:** 1National Laboratory of Nanotechnology (LANOTEC), National Center for High Technology (CeNAT), 1174-1200 San José, Costa Rica; luis.castillohenriquez@ucr.ac.cr; 2Physical Chemistry Laboratory, Faculty of Pharmacy, University of Costa Rica, 11501-2060 San José, Costa Rica; 3Laboratory of Pharmaceutical Technology, Faculty of Pharmacy, University of Costa Rica, 11501-2060 San José, Costa Rica; castro2593@hotmail.com; 4Nuclear Research Center, Faculty of Science, Universidad de la República (UdelaR), 11300 Montevideo, Uruguay; mary@cin.edu.uy; 5Laboratory of Polymers (POLIUNA), Chemistry School, National University of Costa Rica, 86-3000 Heredia, Costa Rica

**Keywords:** critical solution temperature, inflammation, nanomedicine, smart polymers, tissue engineering

## Abstract

Innate and adaptive immune responses lead to wound healing by regulating a complex series of events promoting cellular cross-talk. An inflammatory response is presented with its characteristic clinical symptoms: heat, pain, redness, and swelling. Some smart thermo-responsive polymers like chitosan, polyvinylpyrrolidone, alginate, and poly(ε-caprolactone) can be used to create biocompatible and biodegradable scaffolds. These processed thermo-responsive biomaterials possess 3D architectures similar to human structures, providing physical support for cell growth and tissue regeneration. Furthermore, these structures are used as novel drug delivery systems. Locally heated tumors above the polymer lower the critical solution temperature and can induce its conversion into a hydrophobic form by an entropy-driven process, enhancing drug release. When the thermal stimulus is gone, drug release is reduced due to the swelling of the material. As a result, these systems can contribute to the wound healing process in accelerating tissue healing, avoiding large scar tissue, regulating the inflammatory response, and protecting from bacterial infections. This paper integrates the relevant reported contributions of bioengineered scaffolds composed of smart thermo-responsive polymers for drug delivery applications in wound healing. Therefore, we present a comprehensive review that aims to demonstrate these systems’ capacity to provide spatially and temporally controlled release strategies for one or more drugs used in wound healing. In this sense, the novel manufacturing techniques of 3D printing and electrospinning are explored for the tuning of their physicochemical properties to adjust therapies according to patient convenience and reduce drug toxicity and side effects.

## 1. Introduction

Scaffolds are biocompatible and biodegradable support structures that reproduce an extracellular matrix (ECM) environment. The tissue is grown outside the body to mimic a biological process or to replace a damaged body’s tissue [1,2]. Regarding that, tissue engineering, first introduced by Langer and Vacanti in 1993, aims to employ these structures for different biomedical applications that restore, maintain, and improve damaged tissue functions [3]. This multidisciplinary field analyzes the requirements of the biomaterials needed to produce the scaffolds, such as morphology, mechanical, and surface properties [4,5].

Wound healing is of great interest for tissue engineering. It involves hemostasis, inflammation, proliferation, and remodeling. Each stage comprises different necessary biochemical mediators for a successful process [6]. Here, scaffolds represent excellent structures for wound healing due to their capacity for tissue regeneration and cell growth. In addition, they can perform as drug delivery systems when composed of smart polymers that respond to certain stimuli (e.g., pH, temperature, magnetic, and electric fields) [7,8,9]. Currently, polymer therapeutics is a major interest in the nanomedicine field for the development of novel drug delivery systems [10,11,12,13,14].

Thermo-responsive polymers are very useful for scaffold development due to their outstanding performance under a determined change in temperature (e.g., locally heated tumors in inflammation) [15,16]. This change can induce a phase transformation in the polymer, causing the release of a loaded anti-inflammatory, antimicrobial, and/or wound care drug. Heskins et al. were pioneer scientists who worked with a thermo-responsive polymer, that is, poly(N-isopropyl acrylamide) (PNIPAAm) [17]. According to Xu et al., several other research groups have used it for different biomedical applications. In addition, it has been used in combination with other polymer blocks to improve physical properties as well as biocompatibility and biodegradability [18].

Other research, such as that reported by Jyoti et al., focused on burn wounds for which chitosan has been widely employed as an assistant for fusidic acid [19]. Benchaprathanphorn et al. also tested poly(N-isopropryl acrylamide-*co*-acrylamide) (PNIPAAm-*co*-Am) to fabricate a keratinocyte-fibroblast tissue for treating burn wounds, which showed cell migration and an organization similar to skin tissue’s structure [20]. Both approaches are considered as positive and improve wound healing rate.

Moreover, different novel manufacturing techniques are widely been employed such as 3D bioprinting and electrospinning [21,22]. Kurakula and Rao highlighted the relevance of polyvinyl pyrrolidone (PVP) as a polymer with versatile properties that allow its use for the mentioned fabrication techniques [23]. Aside from that, Nun et al. describe the advantages provided by 3D printed scaffolds and electrospun nanofibers as replaceable wound dressings. The previous work provides useful criteria for designing scaffold architecture and polymer composition for an adequate wound healing process [24]. Here, we present a comprehensive and integrative update of thermo-responsive polymers used for the development of bioengineered scaffolds with drug delivery applications in wound healing. The work is based on the main findings of 233 papers published between 2010 and 2020. The literature search was conducted in Science Direct, Pub Med, and Scopus databases. Therefore, this review aims to demonstrate the magnificent capacity of bioengineered scaffolds to provide spatially and temporally controlled drug release in wound healing, while providing a platform for tissue regeneration. In addition, the novel manufacturing techniques of 3D printing and electrospinning are explored for their creation and tuning of their physicochemical properties.

## 2. Immune Response in Wounds

The immune system possesses a critical role in discriminating harmful pathogens from the body’s healthy tissues. Although it must generate an adequate response to eliminate any strange object, it also has to avoid self-tissue damaging to allow a proper wound healing process [25]. In order to accomplish that, immunity is based on two components: the innate and adaptive responses. The first one takes immediate action upon the detection of an invader, while the second one requires the activation of the innate [26,27]. However, there is evidence that the innate response can be influenced by the adaptive as well. The previous has been explained by some cells exhibiting functional properties of both, such as dendritic cells, gamma delta (+) T lymphocytes, and Langerhans cells [28,29].

Moreover, the immune response in wound healing is a complex process to return the system to homeostasis involving cellular and biochemical mediators in response to a tissue injury caused by trauma, microbes, or foreign materials. Consequently, a series of events including coagulation, inflammation, epithelization, proliferation, and remodeling take place leading to wound closure [30,31,32,33,34]. However, this section aims to provide an overview of the topic, so the attention will be paid to inflammation as it provides the micro-environmental conditions that are necessary for thermo-responsive drug delivery of wound healing substances through bioengineered scaffolds.

The inflammatory process is an early required phase for wound healing (Figure 1), characterized by five typical symptoms: redness, swelling, heat, pain, and loss of tissue function [35,36,37]. Endothelial cells express cell adhesion molecules that promote the binding of circulating leucocytes. Moreover, neutrophils are the first inflammatory cells arriving at the injury site, responding to chemokines, and being chemo-attracted by C5a and C3a complement activation fragments [38,39]. In addition, platelet aggregation and macrophages degranulation trigger the release of other proinflammatory cytokines such as tumor necrosis factor-α, interleukin-1 (IL-1), IL-6, and growth factors such as the transforming growth factor-beta (TGF-β). As fewer proinflammatory substances are released and more pro-regenerative mediators are produced, inflammation is reduced and damaged tissues are repaired [40,41].

## 3. Thermo-Responsive Smart Polymers

In general, water-soluble smart polymers change their physicochemical properties upon the influence of an external stimulus, and some of them are responsive to multiple stimuli. This modification is related to their arrangement, solubility, or the hydrophilic–hydrophobic balance [42,43,44]. Regarding the thermo-responsive polymers, these have been thoroughly studied and exhibit a volume phase transition at a critical solution temperature (i.e., the temperature where exists a balance in the competition established by hydrophilic and hydrophobic chains), usually referred to as cloud point (T_cp_), which is responsible for the changes in the solvation state [45,46,47]. Topical applications and injectable biodegradable scaffolds made of this type of polymers make use of body temperature to cause a change in the physical properties of the system [48].

According to their origin, these polymers can be classified as natural, synthetic, and hybrid. Natural polymers such as chitosan, gelatin, collagen, and cellulose have been widely used for biomedical applications as ECM due to their great biocompatibility and bioactivity. However, their main limitations are related to batch variability and unsuitable physicochemical properties for certain manufacturing processes [49,50,51,52].

On the other hand, synthetic polymers such as PNIPAAm, poly(lactic acid) (PLA), poly(ε-caprolactone) (PCL), poly(N-vinyl caprolactam) (PNVCL), polyethylene glycol (PEG), and polyethylene oxide (PEO) provide greater tunability of their properties and outstanding mechanical behavior that allows using them for different processing techniques. Nevertheless, these polymers may not present the same biodegradable performance as the natural, as well as exhibit lower biocompatibility [53,54]. Remarkably, the limitations exhibited by natural and synthetic polymers can be overcome by their blending, obtaining a hybrid polymer [55,56,57].

### Phase Transition Thermodynamics and Critical Solution Temperature

Polymer solubility is a complex process that depends on their structure and molecular weight, as well as on the viscosity of the system [58]. Based on the Gibbs–Helmholtz equation (ΔG = ΔH − TΔS), changes in Gibbs free energy of the system (ΔG) to negative values represent the condition under which polymers are soluble [59]. This happens when the change in entropy (ΔS) increases due to the diffusion of solvent molecules through the polymer, where polymer–solvent interactions break intermolecular polymeric bonds [60]. In this sense, an adequate solvent can expand polymer molecules, thus decreasing ΔG, while a poor one causes them to collapse. However, the Flory–Huggins solution theory should be addressed in explaining the temperature’s influence on polymer–solvent, polymer–polymer, and solvent–solvent interactions [61,62,63].

Thermo-responsive polymers possess a unique property of solid–gel transition above a certain temperature, and some of them suffer this phase transition near 37 °C, which is the physiological human body temperature (i.e., normothermia). Furthermore, they can be modified to exhibit that change at the desired temperature [61,64,65]. These polymers are classified according to their critical solution temperature in lower critical solution temperature (LCST) or an upper critical solution temperature (UCST) [66,67]. Figure 2 shows a phase diagram where LCST and UCST are represented as solid curves with a single-phase region in between. When the system exhibits a positive ΔG at a certain temperature, the polymer will not be miscible under those conditions, and two different phases will coexist [68,69].

In the first place, polymers exhibiting LCST (usually close to normothermia) are completely miscible in aqueous systems below that parameter as ΔG is negative [70]. The previous is due to the negative change in enthalpy (ΔH) for the dissolution process caused by water molecules surrounding the hydrophilic part of the polymer [71]. In addition, the formation of a structured water molecule arrangement around the hydrophobic part of the polymer provides a negative ΔS [70]. However, above the LCST these substances experience a reversible phase transition from a hydrophilic configuration to a dehydrated or hydrophobic state. Heating induces that transition under an entropy-driven process caused by the loss of ordered water molecule arrangement around the hydrophobic polymer chain [72,73].

Phase separation in LCST polymers is influenced by the interruption in polymer–water hydrogen bonding and the increment in hydrophobic interactions in the polymer chain due to further increase in temperature. When the positive overall ΔS overcomes the negative ΔH, it gives ΔG a positive value that results in chain collapse and a decrease of solubility (Figure 3) [74,75,76]. These materials are usually referred to as negative temperature-sensitive polymers, and great interest has been paid in their coil-to-globule conformational transition in aqueous systems [72].

On the other hand, solubility and physical changes of some polymers are due to UCST. Above that parameter, ΔS and ΔH decrease with the increase in temperature, showing the opposite behavior to that shown by LCST polymers, and thus these materials remain miscible in solution [78,79]. Nevertheless, a phase separation governed by the enthalpy of the system occurs at temperatures below the UCST due to the balance between intra- and intermolecular forces, as well as solvation changes [80]. These materials are also called positive temperature-sensitive polymers and are based on a combination of acrylamide (AAm) and acrylic acid (AAc) [81].

Moreover, some systems can exhibit both behaviors as shown in Figure 2, where an hourglass-shaped phase diagram shows the overlap of each set of curves. When that happens, phase separation is so well defined that the intermediate region is immiscible. In these cases, the temperature range between LCST and UCST tend to be sensitive to the polymer molecular weight and changes in pressure [82,83,84]. Although thermo-responsive systems under an aqueous environment are of great interest for biomedical applications, it is not usual to see both, LCST and UCST behaviors when using them for that purpose. Furthermore, they are not restricted from using other solvents for additional applications [85,86].

## 4. Bioengineered Thermo-Responsive Scaffolds

Scaffolds provide templates for tissue regeneration and physical support for cell growth [87]. These can be made of artificial or natural thermo-responsive polymers, which can condition the different biomedical applications due to their effect on the functional attributes [55,88]. This type of smart polymers has been widely used as a scaffold in non-invasive methods for different tissues, such as skin and heart [89,90]. The previous is attributed to their injectability and self-healing properties, but also their porosity has been highlighted as an outstanding property, which provides enough space for cell migration and tissue vascularization [91].

Moreover, when these polymers are used for the creation of bioengineered scaffolds for wound healing, they must provide a 3D architecture according to the structural heterogeneity of the host tissue environment [92,93]. The previous allows improving the mechanical and cellular activity (e.g., adhesion and proliferation) required by these structures [94,95,96]. In addition, scaffold design needs to consider several features such as cell–tissue interaction; vascularization; scaffold degradation; and loading with drugs, growth factors, cells, and antibacterial material. Therefore, preformulation and rational designs of scaffolds for drug delivery systems or biomedical devices are crucial for developing a functional, biocompatible, and non-immunogenic product of quality that accelerates local tissue healing [97,98].

### 4.1. Novel Manufacturing Techniques

Scaffolds’ relevance lies in their design as bioactive systems than mere cells or drug carriers. Some fabrication techniques provide surface modification, while others take advantage of their physiological thermo-responsive behavior for creating structures with particular and unique geometries. The ability to design a system that can respond to an external stimulus, controlling their degradation, drug release, and healing capacity yield special interest in the development of scaffolds [99,100]. A brief overview of some novel techniques is presented below.

#### 4.1.1. 3D Printing

3D printing or additive manufacturing is a process controlled by a computer that transforms the digital model data created by computer-aided design (CAD), causing sequential layers to deposit on top of each other for creating different geometric structures [101,102]. The obtained 3D objects are highly customized and represent a cost-efficient production. This technique is probably the most adequate for controlling and modifying the internal microarchitecture of scaffolds [103,104,105].

However, not all thermo-responsive polymers are easily employed for 3D printing. Some natural polymers need to be modified or blended with other polymers in order to acquire the rheological and mechanical specifications [106,107]. Biomaterials need to fulfill the requirements of printability, mechanical strength, and degradation behavior to be subjected to this tissue engineering technique. Regarding that, printability determines the capacity of a construct to imitate the 3D structure of biological tissues [108,109].

Fused Deposition Modeling (FDM) is being explored for processing polymers into drug delivery systems required by special populations for treating rare diseases. The process consists of a plastic filament that is pushed into a heating head (Figure 4) [110]. The contact causes the filament to soften into a semi-solid state that forms the sequential layers by extrusion. The heating head or a bed moves in three dimensions, which allows the layers to deposit with great precision [111]. This extrusion method is widely employed for thermo-responsive polymers allowing larger constructs than other alternatives [112].

Another technique, stereolithography, is a laser-based 3D printing technology that makes use of UV-sensitive liquid resins for fabricating the 3D objects [113]. PNIPAAm has been employed for creating thermo-responsive hydrogels that can be used in drug delivery applications [114]. On the other hand, inkjet printing has been used to create highly porous structures. Material’s particle size must be greater than 1 μm, its viscosity has to be lower than 20 cP, and possess a surface tension around 50 mN/m [113].

Fischetti et al. employed inkjet printing, where chitosan was blended with gelatin to form a polyelectrolyte complex to improve printability for the fabrication of scaffolds for anisotropic tissues (e.g., skin andskeletal muscle). The printing temperature was set below the LCST of the polymer blend. Tripolyphosphate was used as a crosslinker for the creation of the scaffold, which greatly conditioned its mechanical properties. The scaffold showed cytocompatibility to L929 cells, and its stability was related to the content of gelatin [115].

Furthermore, synthetic materials are also employed, offering a better resolution for the bioprinting of scaffolds due to the ease of tunability. Seyednejad et al. developed a 3D scaffold base on hydroxyl-functionalized polyester (poly (hydroxymethylglycolide-co-ε-caprolactone) (PHMGCL). The structure showed enhanced hydrophilicity, higher degradation rate, and improved cell support than a PCL 3D scaffold, representing a great template for tissue engineering [116].

#### 4.1.2. Electrospinning

This polymer processing technology allows obtaining nanofibers with high surface-to-volume ratio, highly porous structures, and diverse morphologies that can be easily controlled through different methods such as melt, emulsion, coaxial, multi-jet, side-by-side, and co-electrospinning [117,118].

Figure 5 shows the basic set-up, which consists of a feeding unit with a spinneret that transports the polymer solution until its tip, usually controlled by a pump [119]. A high-voltage supplier is connected to the spinneret and the collector, charging them oppositely. Once the voltage is applied, the polymer’s solution droplet at the tip elongates forming the Taylor Cone. Finally, when the electric field reaches a critical voltage, it overcomes the polymer solution surface tension and causes a charged liquid jet to move downfield until it is deposited in the collector, forming the nanofibers [120,121,122].

Electrospun nanofibers are of great interest to the biomedical and bioengineering industry due to their outstanding properties in terms of biocompatibility, biodegradability, and high drug loading capacity to perform as drug delivery systems [119,123]. Regarding that, these nanofibers can be employed for the fabrication of scaffolds for wound healing that provide either an immediate or controlled release of the active pharmaceutical ingredient (API).

Therefore, electrospun nanofibers composed of thermo-responsive polymers offer a novel solution to current drug delivery inconveniences for wound healing due to their safety profile [124,125,126,127]. However, not all polymers can be employed for this technique asthey need to be soluble in a certain solvent [128,129]. Meng et al. fabricated a poly(lactic-*co*-glycolic acid) (PLGA)/chitosan nanofibrous scaffold by electrospinning. The nanofibers exhibited biocompatibility and biodegradability, as well as a higher drug release with increasing concentrations of chitosan [130].

In another approach, Ji et al. fabricated a PCL-based nanofibrous scaffold and loaded the model protein bovine serum albumin (BSA) through coaxial and blend electrospinning. The coaxial electrospun nanofibers showed uniform morphology with a core–shell structure, while the blend nanofibers possessed defects on their surface and heterogeneous protein distribution. Regarding their release profile, the coaxial scaffold demonstrated a sustained release and provided more protection to the BSA. Therefore, this work demonstrated how different methods can tune-up scaffold properties according to the manufacturing technique [131].

### 4.2. Biocompatibility and Biodegradability

The ECM is a complex and bioactive scaffold constituted mainly by collagen, as well as other proteins and polysaccharides. These components association define their arrangement, which varies from tissue to tissue and defines its mechanical and structural properties. As a result, ECM is a hard-to-replicate structure [132]. However, new generations of thermo-responsive polymers offer the opportunity to synthesize them controlling their architecture and microstructure, thus providing great advances in tissue engineering and drug delivery [133,134]. Their use in the development of bioengineered scaffolds must provide cell support and protection during the healing process, as well as facilitate the deposition method [135]. However, these biomaterials properties (e.g., size, shape, surface area, roughness, and chemical composition) influence the host response, causing variations in the intensity and duration of the inflammatory and wound healing processes. These define the biocompatibility of the polymers and scaffolds [136].

Biocompatibility is the ability of an introduced material into a physiological environment to perform as intended without inducing an inappropriate micro-and macroscopic host response [137]. Implanted scaffolds can activate the immune response, which as explained earlier in this review, involves a series of proinflammatory biochemical molecules that trigger the inflammatory process [138]. Precisely, inflammation is a common indicator for determining the host response to a biomaterial, and it needs to be followed up closely to avoid tissue damage [139,140]. Besides, the presence of massive fibroblast proliferation with associated collagen deposition represents a biocompatibility issue causing extensive scar tissue and fibrous encapsulation [141].

Biomaterials can coexist with specific tissues and physiological systems. A critical aspect lies in selecting the proper biomaterial for the development of the desired scaffold taking into consideration the possible cellular interactions [142]. Research groups like Ratanavaraporn et al. have been working on biomaterials with the ability to locally suppress proinflammatory cytokines during tissue regeneration [143]. The previous allows a longer coexistence between the biomaterial and tissues. Moreover, as stated by Gonzalez et al., a novel advance in this field is the implementation of biomimetic strategies such as the mechanisms from viruses and bacteria, which provide immune system evasion [144,145,146].

Polymers, like any other biomaterial, need to fulfill certain criteria to be used for tissue reparation and wound healing. In general, they must be water-soluble, non-toxic, non-immunogenic, and safe during the whole process including the excretion (i.e., the size below the renal threshold) [147,148]. When used for drug delivery applications, they have to work as drug carriers, reducing the degradation of the API. Furthermore, they should provide a biodegradable character to the scaffolds asthese are not intended as permanent within the body [149,150]. However, their degradation can generate particles that may stimulate an inflammatory response or produce toxic effects. In this sense, the degradation mechanism, kinetics, and its intermediate products have to be taken into consideration, as well as the scaffold’s porosity that is directly linked to the degradation process [151,152,153].

Cho et al. evaluated cell biocompatibility in a hydrophilic PCL/PVP-b-PCL electrospun nanofiber-based scaffold. The authors highlighted the importance of the ECM hydrophilicity as a factor affecting cell adhesion in tissue engineering, and more specifically in PCL. Therefore, they enhanced its surface hydrophilicity through electrospinning with the biocompatible PVP-b-PCL block copolymer. It was reported an increase in the hydrophilic character of the nanofibers as the concentration of PVP-b-PCL block copolymer was raised. Furthermore, the scaffolds exhibited no cytotoxicity, enhanced cell adhesion, and improved viability of primary fibroblasts compared with the initial PCL scaffolds [154].

In another electrospinning approach, Ji et al. evaluated the effect of nano-apatitic particles (nAp) on the biocompatibility and biodegradability behavior of 3:1 polymeric electrospun PLGA/PCL-based scaffolds. The research group prepared nanofibers with 0–30 wt% of nAp that were subcutaneously implanted in rats after their creation and following a 3-week pre-degraded status in order to evaluate in vivo tissue response. The study reported a delayed polymer degradation dependent on nAp concentration. In terms of biocompatibility, nAp significantly improved the tissue response during 4-week implantation; thus, their results are considered as effective for controlling the in vivo adverse reaction of PLGA materials [155].

A study conducted by Xu et al. presented a novel method for 3D-printing of nanocellulose hydrogel scaffolds. The printed scaffolds from a 1 wt% nanocellulose hydrogel supported fibroblasts proliferation as well as exhibited suitable biocompatibility and biodegradability behaviors [156]. In another study, Intini et al. developed a 3D-printed chitosan-based scaffold for wound healing in diabetes. They evaluated the biocompatibility and toxicity toward human fibroblasts and keratinocytes, reporting significant in vitro cell growth. In addition, the in vivo evaluation of the 3D-printed scaffolds in diabetic rats showed an improvement in the restored tissue compared to a commercial patch [157].

Li et al. developed biodegradable soy protein isolate-based waterborne polyurethane composite (SWPU). The biocompatibility and biodegradability of the composite with soy protein isolate (SPI) content of 0%, 10%, 30%, 50%, and 70% were evaluated by in vivo implantation using Sprague Dawley rats. The histological evaluation at 1, 3, 5, 7, and 9 months revealed a slight inflammatory reaction but when there was present SPI this inflammation decreased as time passed. Besides, at 7 and 9 months, the analyses revealed cell differentiation and maturation in granulation tissues for the groups SWPU-30%, SWPU-50%, and SWPU-70%. Moreover, SPI influenced the degradation rate, where SWPU-70% was almost completely degraded in the fifth month, while SWUP-30% was still significantly present in the ninth month [158].

In another work published by Ichanti et al., agarose/collagen-based composited hydrogels presented an outstanding capacity to be used as scaffolds for tissue engineering. The immobilization of collagen in agarose hydrogel not only supports the formation of a multi-layered extracellular network but improves the biocompatibility of this system and its bioactivity as well [159]. Furthermore, a study by Abbaszadeh et al. reported the synthesis of a novel chitosan-based quercetin nanohydrogel, which besides its antibacterial and anticancer activity, its biocompatible behavior makes it a novel alternative for different purposes such as wound healing [160].

### 4.3. Biopharmaceutical Enhancement

Low drug bioavailability is regarded as one of the major challenges for pharmaceutical industries [161]. Many of the newly discovered APIs are classified within the Biopharmaceutical Classification System (BCS) (i.e., a classification system according to the drug’s aqueous solubility and intestinal permeability) as class II or IV, which means that they possess poor water solubility [162,163]. Some photochemicals obtained through plant extract also exhibit low solubility. This brings difficult challenges for the development of drug delivery systems [164]. On the other hand, only a few new drugs are considered as class I or III, that is, when the maximum clinical dose is dissolved in 250 mL of water in a pH range from 1.2 to 6.8 [165].

Moreover, the growing demand for personalized therapies is being addressed by the nanomedicine field. The use of thermo-responsive polymers can help overcome the obstacles for successful drug delivery. The incorporation of these biomaterials in scaffolds can enhance the drug’s low solubility, bioavailability, protect from enzymatic degradation, provide fast clearance rates, and they cannot cross biological barriers [77,166,167]. Zhao et al. enhanced piroxicam solubility in water by evaluating different mixtures with gelatin. The study reported that the formulation composed of piroxicam/gelatin 1:8 released about 85% of the loaded drug after 10 min. This result suggests the developed system as a promising strategy for improving the biopharmaceutical performance of the API [168].

Furthermore, a drug’s high crystallinity is a cause of poor dissolution rate, and thus low bioavailability [161]. Therefore, strict control of the crystalline state during the manufacturing and the use of solid soluble forms such as amorphous or anhydrous systems can improve greatly this issue. However, polymers like PVP and hydroxypropyl methylcellulose (HPMC) can also inhibit crystallization [169]. Luo et al. worked with the hydrophobic drug Tanshinone (i.e., traditional Chinese medicine from *Salvia miltiorrhiza Bunge*). The research group found that its crystallinity could explain the exhibited poorly water solubility. They evaluated the biocompatible sodium alginate for suppressing the drug crystal growth. The analyses revealed a reduction in crystalline behavior, which caused an improvement in the dissolution rate, bioavailability, as well as in pharmacological activity [170].

The manufacturing process employed for scaffolds production can also improve drug physicochemical and biopharmaceutical properties that influence bioavailability [171]. Electrospinning improves the drug’s solubility through the amorphization of the API and the nanofiber’s high surface-to-volume ratio [119]. Llorens et al. made use of this technology to successfully develop triclosan-loaded PLA/PEG scaffolds with tunable hydrophilicity and porosity [172]. The same research group also loaded triclosan and curcumin in a scaffold constituted by PEG/poly(butylene succinate) (PBS) fabricated through coaxial electrospinning. Asit was not possible to solubilize PEG in the aqueous medium, the incorporation of the hydrophobic drugs occurred [173].

Furthermore, 3D printing can create unconventional structures with complex geometries that possess the ability to incorporate poorly water-soluble drugs and provide personalized medicines [174,175]. Although their low surface area has limited their drug loading capacity, Dang et al. created a 3D printed a dual macro-, microscale porous PCL network to easily and efficiently load different drugs [176].

## 5. Drug Delivery Applications of Bioengineered Thermo-Responsive Scaffolds in Wound Healing

Scaffolds’ behavior and mechanism are highly influenced by the physicochemical properties of the thermo-responsive polymers used for their development but also due to the regulation systems of the biological host. These natural feedback (e.g., inflammation, hyperthermia) aims to stabilize any condition that contrasts with the physiological balance [97]. As a result, scaffolds and their constituent biomaterials make use of these biological responses to provide novel tools for drug delivery. These systems can be applied to the wound healing process; accelerating tissue healing, cicatrization process, and regulating the inflammatory response [177].

Scaffolds made of synthetic, natural, and modified biopolymers are loaded with small drugs or biomacromolecules (e.g., proteins and poly(nucleic acids)) [178,179]. For instance, polymers exhibiting the nonlinear LCST behavior are the ones employed for wound healing drug delivery [180]. As mentioned before, these systems suffer solubility alterations upon an increase in temperature, usually above the normothermia, where a reversible transition from a hydrophilic to a hydrophobic state takes place. Drug release is reduced below their LCST, and is carried out mainly by surface desorption, swelling, and degradation of the polymer matrix. For high-swelling hydrophilic forms, the release depends on the diffusion through the polymer matrix, while for low-swelling polymers it is subjected to the swelling process itself [181,182].

Many biopolymers with a biodegradable nature can perform as described previously such as PCL, chitin, silk fibroin, and PLA [183,184,185,186,187]. However, the use of chitosan for controlled drug delivery has been of great interest in nanomedicine research [188]. Zamora et al. easily obtained it from the enzymatic deacetylation of chitin present in crustacean shells and tilapia skin [189]. Other remarkable properties such as biocompatibility, biodegradability, and low toxicity make it a great candidate for wound healing. In addition, this polymer is frequently associated with other substances to improve its mechanical and physicochemical properties [188].

Rusu et al. combined chitosan with poly(aspartic acid) to develop a formulation for wound healing. They evaluated different compositions and loaded them with amoxicillin. Drug release was tested for 300 min under 37 °C, and at pH 5.4 and 7.4 aswound healing shows an initial basic environmental pH that shifts to acidic, around 5.5. Drug release from prototypes 0.25NG0.1, 0.25NG0.2, and 0.25NG0.3 were, respectively, 99.1%, 96.5%, and 70.6% at pH 7.4, and 71.5%, 61.2%, and 52.8% at pH 5.4. The three prototypes revealed a controlled release dependent on the composition and performed a sustained release under the acid pH of the wound environment. Therefore, this scaffold is a suitable option for wound bacterial infections and possesses an adequate in vivo biocompatibility [190].

PVP has also attracted the attention of many research groups for developing novel drug delivery systems. Although it is a hydrophilic polymer it can encapsulate either hydrophilic or poorly water-soluble drugs [191]. Its swelling properties and pore-forming capacity were used by Zuo et al. to improve the performance of the developed chitosan/polyurethane scaffold for skin tissue engineering [192]. Moreover, its versatile properties make it an excellent polymer for the fabrication of scaffolds and other biomedical products through electrospinning and 3D-printing [23,193].

Alginate has also been employed for developing novel dressings for wound healing. This polymer has outstanding properties such as biocompatibility, biodegradability, high swelling, porous structure, and can be functionalized [194]. In addition, this polymer has contributed to reducing bacterial infections, less scarring, and promoting cell proliferation [195,196]. Buyana et al. developed a scaffold composed of sodium alginate and Pluronic F127 that was loaded with norfloxacin, zinc oxide (ZnO) nanoparticles (NPs), thymol, and the antifibrinolytic agent, aminocaproic acid. The formulation revealed a synergistic antibacterial effect and helped to regulate clotting. Therefore, this approach represents a potential wound dressing for infected and bleeding wounds [197].

Moreover, thermo-responsive scaffolds provide spatially and temporally controlled drug release strategies for one or more API. These structures can extend drug release avoiding the burst effect as possible. This is a relevant aspect in order to maintain therapeutic levels in the body and reduce the required dose [166,198]. More important, these systems allow achieving a desirable controlled drug release by releasing the API only under the influence of a thermal stimulus. A locally heated tumor presented during inflammation, either caused by tissue damage or as a response upon the introduction of a biomaterial, allows enhancing the release due to polymer chains shrinking [199].

Chen et al. developed a scaffold prototype for tissue engineering with local sustained drug delivery. They firstly created a macroporous PCL base scaffold that was then embedded with a porous matrix consisting of chitosan and nanoclay. This biomaterial allows tuning of drug release rates by adjusting the amounts and types of chitosan in the formulation [200]. Similar research was carried out by Asadian et al., where a scaffold composed of chitosan, poly(acrylic acid), and nano-hydroxyapatite (n-HAP) was loaded with the anti-inflammatory drug naproxen. The maximum loading capacity exhibited by the different prototypes was 34.8%, which corresponded to V4 formulation (i.e., the major concentration of n-HAP). Drug release showed an initial burst during 24 h in all prototypes with the V4 sample showing the best results, maintaining 58.2% of the loaded drug after 14 days [201].

In another approach, Chogan et al. developed a scaffold to treat the major complication in wound healing: fibrosis. In order to address that issue, a three-layer scaffold was designed with a PCL-chitosan layer on the sides and a polyvinyl alcohol (PVA)-metformin HCl in the middle. This development was evaluated in rats and showed a reduction in inflammation, smaller scar area, and an optimal modulation of collagen density after 15 days. Regarding metformin HCl release, the composite scaffold structure reduced the burst release in the initial five hours (35%) and exhibited a linear release profile over the 15-day period. On the other hand, a single-layer PCL could not control the burst release in the initial phase (78%), while a single-layer PVA showed a moderate burst release (38%). Therefore, this development is considered a promising therapeutic approach for metformin HCl delivery in order to reduce scar formation and improving the healing process [202].

Garakani et al. synthesized PLGA microparticles loaded with dexamethasone, which was dispersed in different hydrogels of chitosan/PVP. The obtained scaffolds possessed an amorphous structure that facilitated the dissolution of the microparticles, as well as a high swelling ratio and controlled biodegradability rate. The study reported a slower release upon the addition of PVP. However, the designed scaffolds released 75–85% of the drug after 30 days, while the loaded microparticles fully released the complete dose after 22 days. Therefore, this formulation can be considered as a sustained release thermo-responsive drug delivery alternative for inflammation in wound healing during a 30-day course [203].

In addition, thermo-responsive scaffolds intend to improve patient compliance with therapies. Biswas et al. developed a scaffold through electrospinning for sustained release of the herbal drug Panchavalkala (i.e., a combination of five bark drugs from Ayurvedic medicine). The researchers used PLA as the carrier due to its biodegradable nature. The scaffold released approximately 80% of the drug continuously for 5 days. The developed scaffold proved to have better efficacy in wound healing compared to the traditional unstable dispersion of the drug. Therefore, this biomaterial increased the drug’s bioavailability, thus can effectively control the inflammatory process [204].

Furthermore, thermo-responsive polymers can be used as injectable biomaterials in the form of a hydrogel, as shown in Figure 6 [205]. This allows the in situ formation of scaffolds, minimizing the employment of invasive methods, and representing a novel and advanced drug delivery system especially for subcutaneous application [206,207,208,209]. Hydrogels possess a 3D structure that can be modified in terms of their physicochemical properties to obtain a firmly attached scaffold to the external and internal wound [210]. The technique involves mixing the thermo-responsive polymer with the API at room temperature for subsequent injection into the body. After that, the increase in body’s temperature above polymer LCST induces a phase transition that forms a physical gel, favoring the release of the drug from the scaffold. This matching between the physiological and the gelation temperature represents a great advantage for wound healing [211,212,213].

Andrgie et al. developed an injectable heparin-conjugated PNIPAAm in situ gel-forming polymer with encapsulated ibuprofen to address pain and excessive inflammation during wound healing. In vitro analysis showed a reduction of proinflammatory mediators due to the released drug. In addition, the applied hydrogel on the mice back wound revealed that the formulation improved healing compared to a placebo group, thus presenting this in situ forming-scaffold as a promising therapeutical approach [214].

Dong et al. prepared a novel thermo-sensitive hydrogel composed of PNIPAAm/poly(γ-glutamic acid). The differential scanning calorimetry analysis confirmed the thermo-sensitivity at normothermia and revealed a phase transition temperature at 28.2 °C. This hydrogel exhibited a high swelling rate, good biocompatibility, and higher wound closure rate. In addition, superoxide dismutase was loaded to improve trauma treatment in wound healing due to its antioxidant activity over the reactive oxygen species [215]. Another important regulator in wound healing is nitric oxide (NO), which modulates cell proliferation, wound contraction, collagen deposition, and has antibacterial activity. Regarding this, Cao et al. developed a thermo-responsive hydrogel constituted by S-nitrosoglutathione, pluronic F127, and alginate, in which NO was incorporated for treating infected wounds. The scaffold demonstrated biocompatibility, sustained release of NO for seven days, and bactericidal activity against *Staphylococcus aureus* and *Pseudomonas aeruginosa* [216].

As presented in Table 1, there are several drug delivery applications of thermo-responsive scaffolds for wound healing such as pain, inflammation, diabetes, microbial infections, and prevention of large scar tissue [217,218].

Chronic wounds and ulcers caused by different diseases such as diabetes demand advanced therapies for treating them aschronic inflammation, infections, and poor tissue regeneration are complications that can lead to amputation [234,235]. Lee et al. developed core–shell nanofibrous bioactive insulin-loaded PLGA scaffolds through coaxial electrospinning for sustained release of the synthetic hormone in diabetic rats. The scaffolds exhibited a release of the molecule during four weeks, which promoted diabetic wound healing [236].

Karri et al. explored the application of curcumin in the management of diabetic wound healing. In this study, they developed a novel nanohybrid scaffold that consisted firstly in the incorporation of curcumin in chitosan NPs to a subsequent impregnation into a collagen scaffold, which provides better tissue generation. The study suggests that the synergistic combination of curcumin as an anti-inflammatory drug, and chitosan and collagen as a drug carrier and wound healing scaffold have an outstanding healing capacity in diabetes [237].

Moreover, as reported by Hao et al., thermo-responsive scaffolds have been employed for tissue regeneration and controlling the inflammation caused by periodontal diseases. In this study, a biosensitive PLGA/mesoporous silica nanocarrier core–shell porous microsphere encapsulated PLA spongy nanofibrous micro-scaffold was developed for local injection delivering of celecoxib into periodontal tissue. The drug release provided significant control of the inflammation, while the scaffold contributed to the formation of new tissue, resulting in an effective approach for treating periodontal disease [238].

The study reported by Zehra et al. presents a concern for scar-free healing and pain management in wound healing. To address this, the research group developed a 3D porous biomimetic scaffold with a novel combination of polymers: chitosan and sodium alginate. Additionally, the scaffold was loaded with ibuprofen. The development resulted suitable for tissue engineering applications due to its nano- and microporous structures. Furthermore, the scaffold showed a sustained drug release in vitro, which is considered ideal for the sake of minimal inflammation and pain management [239].

The biofunctionalization of polymeric fibrous scaffolds is being regarded as a novel approach to improve the incorporation of bioactive molecules without affecting their activity and loading capacity. In this sense, Cheng et al. created PLGA electrospun fibrous scaffolds biofunctionalized with PEG, bFGF growth factor, cell adhesive peptide (RGD), and loaded with 20(R)-ginsenoside (Rg3). The synergistic effect between the biofunctionalized scaffold and Rg3 promoted early wound healing in rabbit ear wounds and inhibited the formation of hypertrophic scars [240].

Furthermore, wounds are vulnerable to suffering from bacterial infection, which can extend the inflammatory process and increase its intensity [241,242]. Several research groups have worked on different strategies that combine natural antimicrobial and anti-inflammatory approaches for wound healing [243,244,245]. Regarding this, Garcia et al. developed an electrospun PCL-based anti-inflammatory scaffold loaded with thymol (THY) and tyrosol (TYR) essential oils. The study aimed to reduce inflammation and minimize the risk of infected wounds, as well as reducing antimicrobial resistance due to the indiscriminate use of antibiotics. Furthermore, the authors reported that PCL-THY exhibited a more efficient down-regulation of proinflammatory genes compared to the PCL-TYR and PCL-THY-TYR systems [246].

In another approach, Mahmoud and Salama employed the freeze-drying technique for the preparation of norfloxacin-loaded scaffolds for wound treating. The scaffolds were composed of collagen with chitosan HCl or with chitosan of low molecular weight. Although the selected chitosan conditioned the mechanical strength, both provided an extended biodegradability and showed almost a 100% release of the antibiotic drug after 24 h. Besides, the in vivo study in Albino rats revealed after 28 days of wound dressing that tissue regeneration time was faster compared to non-treated wounds [247].

Moreover, burn infections are also a major concern in wound healing therapies asthey are the most traumatic and physically disabling injuries, leading to high morbidity and mortality rates [248]. In this sense, Lan et al. designed an antibacterial silk fibroin scaffold with gelatin microspheres impregnated with gentamycin sulfate, which were further embedded in the silk fibroin matrix. After 21 days, the scaffold not only served as a tissue regeneration template when evaluated in a rat full-thickness burn infection model, but also provided a sustained release of the API and exhibited stronger antimicrobial activity against *Escherichia coli*, *S. aureus*, and *P. aeruginosa.* Therefore, this can be considered as a promising approach for wound healing and burn infection treatment in severely burned patients [249].

Another research, carried out by Aliakbar et al., explored burn wounds infected with drug-resistant bacteria *Acinetobacterbaumannii* from clinical isolates. The group addressed this challenge by developing a thermo-responsive chitosan hydrogel. The hydrogel antibacterial activity was evaluated in vitro and with a rat model. The in vivo study revealed an accelerated wound healing, re-epithelization, and antibacterial efficacy. In addition, the development showed to be biocompatible and will be further assessed in clinical investigations [250].

## 6. Conclusions

Thermo-responsive polymers are currently one of the most important materials in nanotechnological, tissue engineering, nanomedicine, and biomedical fields for the development of scaffolds. Their amphiphilic nature, the ease for tuning of their physicochemical properties through novel techniques, and the influence of physiological feedbacks enable the delivery of different drugs and biomolecules for wound healing. In addition, these polymers can be injected as a hydrogel and form in situ scaffolds, which minimizes the need for invasive methods. Thermo-responsive scaffolds not only provide physical support for cell growth and tissue regeneration, but also protect the wound site from bacterial infections, allowing better healing to those patients with large burns or diabetes. Moreover, their self-healing properties provide accelerated tissue regeneration and scar-free healing. However, regardless of current advances, further research is encouraged for better control of the release rate of drugs and their burst release at the initial stage. The authors suggest paying special emphasis to the process parameters in order to achieve an optimum design that allows obtaining a high-quality, biocompatible, and biodegradable drug delivery system according to the wound needs.

## Figures and Tables

**Figure 1 ijms-22-01408-f001:**
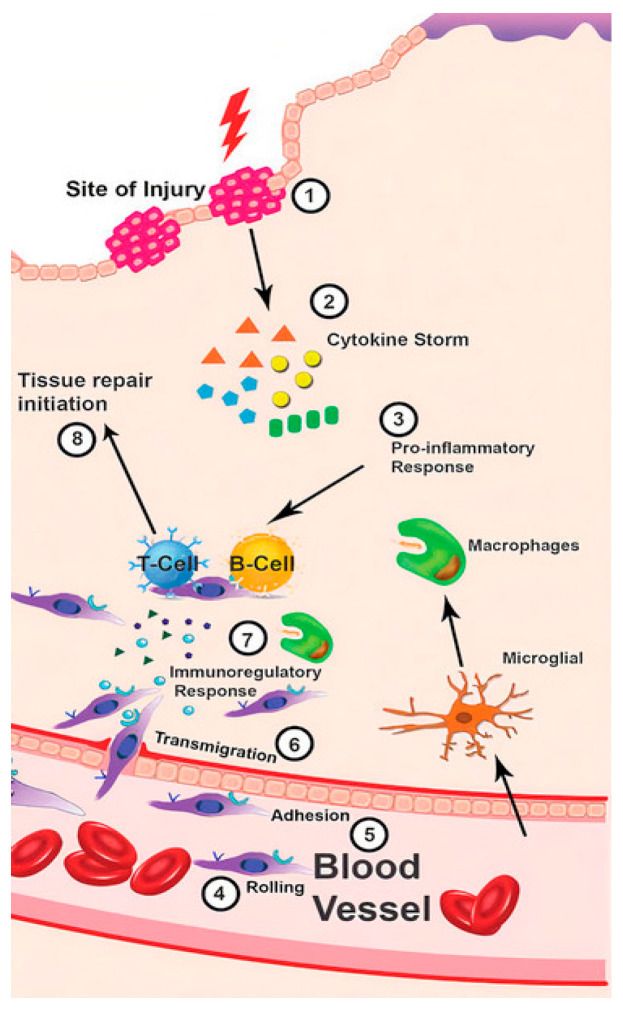
Immune response in wound healing process. Reprinted with permission from Rawat, S. et al. Mesenchymal Stem Cells Modulate the Immune System in Developing Therapeutic Interventions. IntechOpen. Copyright (2019) [35].

**Figure 2 ijms-22-01408-f002:**
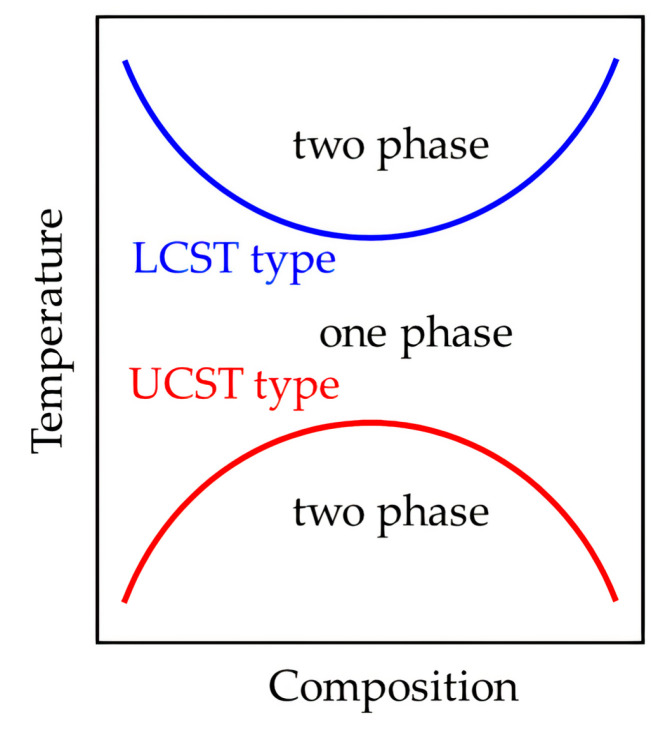
Lower critical solution temperature (LCST) and upper critical solution temperature (UCST) phase transition behaviors of thermo-responsive polymers in solution. Reprinted with permission from Sugeno, K. et al. UCST Type Phase Boundary and Accelerated Crystallization in PTT/PET Blends.Polymers 12(11). Copyright (2020) MDPI [69].

**Figure 3 ijms-22-01408-f003:**
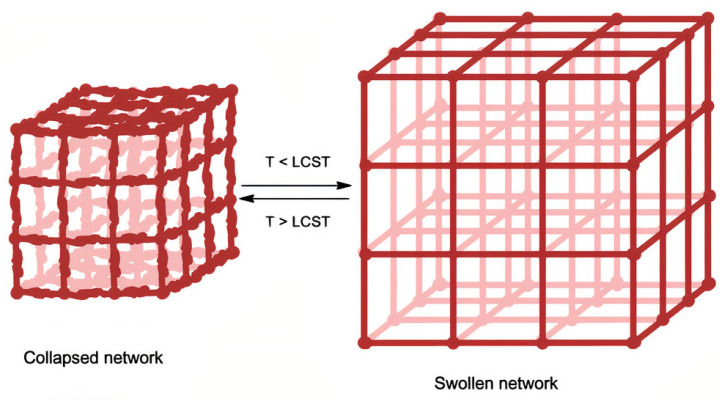
Effect of temperature on the swelling of covalently linked networks in LCST polymers. Reprinted with permission from Ward, M., and Georgiou, T. Thermoresponsive Polymers for Biomedical Applications. Polymers 3(3). Copyright (2011) MDPI [77].

**Figure 4 ijms-22-01408-f004:**
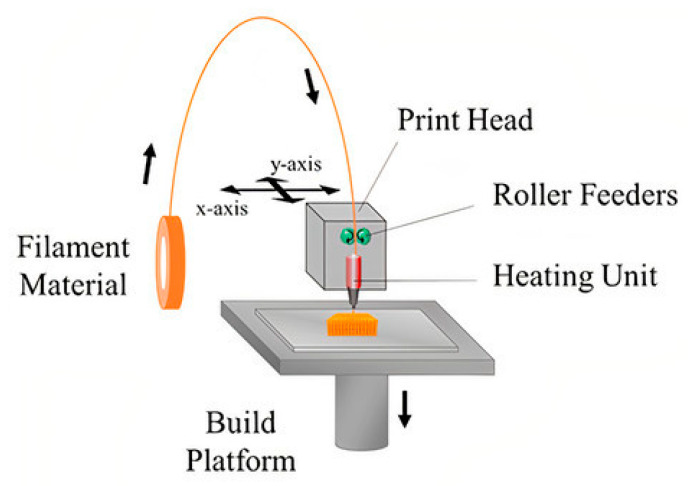
Fused Deposition Modeling. Reprinted with permission from Tamay, D., et al. 3D and 4D Printing of Polymers for Tissue Engineering Applications. Frontiers in Bioengineering and Biotechnology 7. Copyright (2019) Frontiers [110].

**Figure 5 ijms-22-01408-f005:**
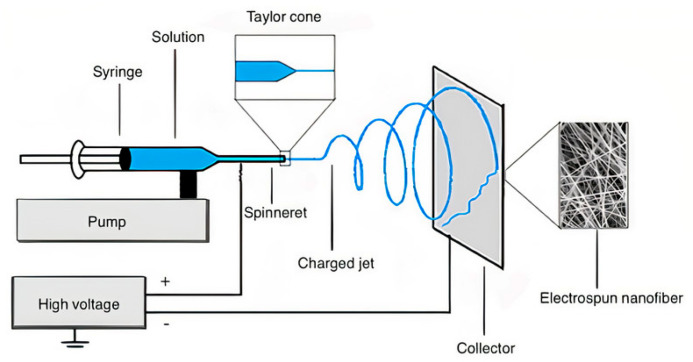
Electrospinning basic set-up. Reprinted with permission from Castillo, L., et al. Electrospunnanofibers: A nanotechnological approach for drug delivery and dissolution optimization in poorly water-soluble drugs. ADMET & DMPK 8(4). Copyright (2020) International Association of Physical Chemists [119].

**Figure 6 ijms-22-01408-f006:**
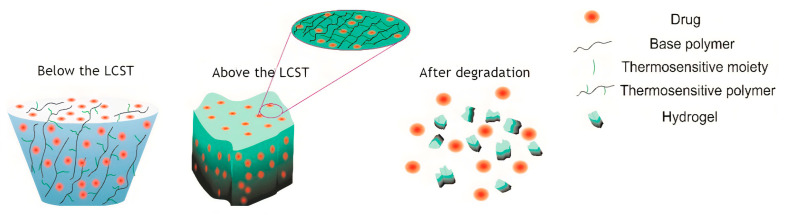
Hydrogel drug delivery system for in situ formation of scaffolds. Reprinted with permission from Ghaeini et al. Thermoresponsive Nanogels Based on Different Polymeric Moieties for Biomedical Applications. Gels 6(3). Copyright (2020) MDPI [205].

**Table 1 ijms-22-01408-t001:** Thermo-responsive scaffolds for drug delivery in wound healing.

Polymer System	Delivered Drug	Application	Release Time	Ref
Gelatin	Ibuprofen	Inflammation and bone regeneration	100 h	[219]
PLGA	Ibuprofen	Inflammation	30 h	[220]
Poly(*N*-vinylcaprolactam-*co*-methacrylic acid)	Ketoprofen	Inflammation	50 h	[221]
Poly(di(ethylene glycol) methyl ether methacrylate), Ethyl cellulose	Ketoprofen	Inflammation	100 h (80%)	[222]
Sodium alginate	Celecoxib	Hyperthermia	-	[223]
Chitosan, PCL	Ferulic acid, resveratrol	Inflammation, pro-angiogenic	120 h (55% of ferulic acid and 48% of resveratrol)	[224]
PVA, chitosan	Tetracycline HCl	Bacterial infection	4 h (80%)	[225]
Chitosan, PEG	Ciprofloxacin HCl	Bacterial infection	20 h (30%)	[226]
Chitosan, alginate	Alpha-tocoferol	Skin injuries, oxidative process	14 days (77%)	[227]
Eudragit	Gentamicin sulphate	Bacterial infection in diabetic ulcer	12 h (90% at acid pH)	[228]
PLGA	Clorhexidine	Infection treatment	50 days	[229]
Chitosan	Vamcomycin	Bacterial infection, osteomyelitis	26 days	[230]
PCL, silk fibroin	Amoxicillin	Bacterial infection	14 days	[231]
PLGA, poly(L-lactic acid)	Doxycycline	Bacterial infection	6 weeks	[232]
PVA	Tetracycline	Bacterial infection	24 h (82% at pH 7.4)	[233]

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
