# Peer review of "Exploration of Bioengineered Scaffolds Composed of Thermo-Responsive Polymers for Drug Delivery in Wound Healing"

_ijms, 2021, doi:10.3390/ijms22031408_

Round 1

Reviewer 1 Report

This is a review paper regarding "Bioengineered Scaffolds for Thermo-responsive Drug Delivery in Wound Healing". However, given that the aim is to write a review article there is not enough citations of the work in this area of research.

The authors need to address the following concerns:

Please clarify- Page 2 "However, there is evidence that one response can be influenced 72 by its counterpart. The previous has been explained by some cells exhibiting functional properties of 73 both, such as dendritic cells, gamma delta (+) T lymphocytes, and Langerhans cells [22,23]". Its not clear what authors want to indicate.

Section 3D printing need elaboration and more references. Need some examples related to the review for thermoresponsive scaffolds and drug delivery for wound healing.

Similarly Section on Electrospinning needs elaboration. 

Section 4.2 the authors described only 2-3 studies, however there is a vast literature in this area which should have been included in the review.

Overall the citied literature is not adequate for a review article and the authors need to include more literature studies and revise their manuscript.

Author Response

This is a review paper regarding "Bioengineered Scaffolds for Thermo-responsive Drug Delivery in Wound Healing". However, given that the aim is to write a review article there is not enough citations of the work in this area of research.

Dear reviewer, we really appreciate your observations. We have corrected our work beginning from the title since the idea of our paper is to present the contribution that the bioengineered scaffolds composed of thermo-responsive polymers have done in terms of drug delivery for wound healing. Therefore, it was also relevant to include a vast literature regarding those smart polymers. We also included relevant references in section 5. “Drug Delivery applications in wound healing”.

Please clarify- Page 2 "However, there is evidence that one response can be influenced 72 by its counterpart. The previous has been explained by some cells exhibiting functional properties of 73 both, such as dendritic cells, gamma delta (+) T lymphocytes, and Langerhans cells [22,23]". Its not clear what authors want to indicate.

Correction made, please check lines 214-217 (with track changes mode) which state the following: “However, there is evidence that the innate response can be influenced by the adaptive as well. The previous has been explained by some cells exhibiting functional properties of both, such as dendritic cells, gamma delta (+) T lymphocytes, and Langerhans cells.”

Section 3D printing need elaboration and more references. Need some examples related to the review for thermoresponsive scaffolds and drug delivery for wound healing. Similarly Section on Electrospinning needs elaboration. 

Section 4.1.1. 3D printing and 4.1.2. Electrospinning have been strengthen with more relevant references, as can be seen in the track changes. However, some relevant works are also discussed in section 4.2. Biocompatibility and biodegradability, 4.3. Biopharmaceutical enhancement and 5. Drug delivery applications in wound healing. We wanted section 4.1. to be an introduction of the techniques.

Section 4.2 the authors described only 2-3 studies, however there is a vast literature in this area which should have been included in the review.

In addition, we have discussed a few other examples in section 4.2. Biocompatibility and biodegradability to illustrate the importance of these factors in the development of a bioengineered scaffold., as can be seen with the track changes mode.

Reviewer 2 Report

The comprehensive submitted is on assessing the how special polymers based bioengineered scaffolds can be modified for thermo-responsive drug delivery and aid in wound healing that is primitive topic but limited number of reviews have been published till date. The review has topics bringing up the importance and need for understanding the material aspects of polymers and their influence on wound healing, signalling pathways & also it rationalises controlled release systems reported to date with various polymers. The review is currently under the scope of the journal and even adds significant information in the respective field. The review is well articulated but further needs more information and modifications as per the following recommendations. The review lacks info graphic or interactive figures those can be in sync with the content. Figures are essential for improved readability.  The authors are even advised for English language revisions as well. These major revisions and additions of relevant latest references are required in order the manuscript to be considered. These major revisions are suggested.

  1. Title seems to be very indirect, It can be “Exploration of the Bioengineered Scaffolds ….. Authors are strongly suggested for this change.
  2. Abstract is too short. Authors need to elevate the importance of understanding the structural aspects or even physical properties – How these thermoresponsive polymers can be promising than available approaches especially in wound healing. Authors are suggested to mention the objective and importance of this review in abstract.
  3. Keywords 4-5 are enough. Lengthy words needs to be replaced. Single words are encouraged.
  4. Introduction has too big sentences need to be short and concise.
  5. Section “Drug delivery applications in wound healing” should be a separate section as 5 not in section 4.
  6. The review seems to be too descriptive- info graphics and figures are encouraged especially extracted from the references cited.
  7. Conclusion seems half informed, authors need to elevate the outlook of the thermo responsive polymers and how newer delivery systems are able to address and limit microbial infections. Process parameters are not collectively discussed in the other sections. Changes are advised.
  8. Thermo-responsive drug delivery in wound healing is a futuristic approach to address the unmet clinical needs. Apart from polymers various biomaterials alone or in combination with other organic components have been explored for wound healing and for treatment of various other ailments. This section is completely ignored in the review submitted. Authors are strongly recommended to include a separate section for the same citing useful and latest references.
  9. The present review touches upon the clinical performance and limitations of all forms of polymers. However, the novel delivery approaches that have been reported for improvement of physiochemical properties, dissolution that have direct effect on bioavailability and in-vivo wound healing performance have been neglected. The authors are strongly recommended to add a new section and address the citing only the latest references not in depth.
  10. Points addressed in 7 & 8 should be touched in the abstract as well.
  11. In support of points raised in 7 & 8, Authors are strongly advised to mention the following latest novel delivery systems prepared using other polymer like PVP, alginates, PCL etc and reported in literature for wound healing. Citing all these below references in introduction and in other relevant separate sections is highly recommended. This could elevate the importance of the review to the readers and support the benefit of this review if accepted or published online.

Polymers 12, no. 11 (2020): 2619.

Int J Pharm Sci Drug Res4(1), 35-43.

Biomacromolecules. 2020 Oct 6.

Journal of Engineered Fibers and Fabrics10(4), 155892501501000411.

Polymers12(3), p.580.

European Polymer Journal (2020): 109919.

Journal of Materials Science: Materials in Medicine 31, no. 12 (2020): 1-12.

International Journal of Pharmaceutics, 119438.

Pharmaceutics12(10), p.926.

Journal of drug delivery science and technology, 102046.

Gels6(3), p.20.

Alginates in Drug Delivery, pp. 323-358. Academic Press, 2020.

Polymers 12, no. 9 (2020): 2010.

 Electrospun Materials and Their Allied Applications, 265-306.

International Journal of Nanomedicine 15 (2020): 7775.

International Journal of Biological Macromolecules. 2020 Oct 15.

Acta Biomaterialia 113 (2020): 177-195

Author Response

  1. Title seems to be very indirect, It can be “Exploration of the Bioengineered Scaffolds ….. Authors are strongly suggested for this change.

Dear reviewer, we really appreciate your observations. We have corrected our work beginning from the title since the idea of our paper is to present the contribution that the bioengineered scaffolds composed of thermo-responsive polymers have done in terms of drug delivery for wound healing. The title of the paper is “Exploration of Bioengineered Scaffolds Composed of Thermo-responsive Polymers for Drug Delivery in Wound Healing”

  1. Abstract is too short. Authors need to elevate the importance of understanding the structural aspects or even physical properties – How these thermoresponsive polymers can be promising than available approaches especially in wound healing. Authors are suggested to mention the objective and importance of this review in abstract.

Since our work pays a special emphasis on the thermo-responsive polymers, we included some relevant information regarding the structural and physicochemical properties of these materials, and relating the physical changes that they undergo due to the physiological response exhibited in the wound healing process. In addition, the objective and importance of the review are stated in lines 29-36 (with the track changes mode).

  1. Keywords 4-5 are enough. Lengthy words needs to be replaced. Single words are encouraged.

We corrected this section leaving only 5 keywords that are not present in the title.. However, some compounded words are left since we consider them to be helpful in facilitating our work to be found by the readers.

  1. Introduction has too big sentences need to be short and concise.

We checked writing and spelling throughout the text. Long sentences were turned into separate ideas.

  1. Section “Drug delivery applications in wound healing” should be a separate section as 5 not in section 4.

We appreciate the observation and followed your advice. Sub section 4.3. Drug delivery applications is now found as section 5.

  1. The review seems to be too descriptive- info graphics and figures are encouraged especially extracted from the references cited.